# The Mediating Role of Sarcopenia in the Association between Physical Activity and Falls among Chinese Older Adults: A Cross-Sectional Study

**DOI:** 10.3390/healthcare11243146

**Published:** 2023-12-12

**Authors:** Chenyu Liang, Lei Shi, Baocheng Li, Zhiyu He

**Affiliations:** Sports Centre, Xi’an Jiaotong University, Xi’an 710049, China; 3121331013@stu.xjtu.edu.cn (C.L.); bcli@xjtu.edu.cn (B.L.); he.zhi.yu@mail.xjtu.edu.cn (Z.H.)

**Keywords:** physical activity, falls, sarcopenia, mediation analysis

## Abstract

Physical inactivity and sarcopenia are potentially modifiable risk factors for falls in older adults, but the strength of the association between physical activity (PA), sarcopenia, and falls in Chinese older adults is unclear. This study sought to investigate the potential mediation mechanism relationship in the connection between PA, sarcopenia, including its elements (muscle strength, physical performance, and skeletal muscle mass), and falls among Chinese older people. The subjects were 3592 community-dwelling Chinese aged 60 or over, selected from the China Health and Retirement Longitudinal Study (CHARLS). PA was evaluated through the International Physical Activity Questionnaire (IPAQ), and sarcopenia was determined through the Asian Working Group on Sarcopenia (AWGS) 2019 guidelines. We employed logistic regression to explore the link between physical activity, sarcopenia, and falls. Additionally, we applied Karlson, Holm and Breen’s (KHB) method to estimate two different mediation models. The results demonstrated that PA lowers the risk of falls [odds ratio (OR) 0.54, 95% confidence interval (CI) 0.48–0.61], whereas sarcopenia increases the risk of falls (OR 1.34, 95% CI 1.16–1.55). Sarcopenia mediated the association between PA and falls, explaining a total of 2.69% of the association (indirect effect = −0.02). PA also had a significant mediating effect on the association between sarcopenia and falls, explaining a total of 20.12% of the association (indirect effect = 0.06). The proportion mediated by sarcopenia was 2.69% for PA and falls (indirect effect = −0.02). Our findings suggest that PA and sarcopenia have a direct effect on falls as well as an indirect effect through each other. Enhancing PA levels and preventing sarcopenia may help prevent falls in older adults.

## 1. Introduction

By 2021, China’s population aged 60 and over will have reached 264 million, accounting for 18.7% of the world’s total population [1]. As the aging process accelerates, the social and economic problems caused by falls among older adults are becoming more prominent [2]. A fall is “an unexpected event in which an older person comes to rest on the ground, floor, or lower level” [3]. In China, approximately one quarter of males and four-fifths of females aged 65 and over have experienced a fall [4]. Falls are linked to physical disability, functional impairment [5], frailty [6], and sarcopenia [7], and have become the predominant factor in physical injuries and fatalities related to injuries among older adults [8]. This has a detrimental effect on seniors’ physical and mental well-being, as well as creating a large financial strain in terms of healthcare costs [9]. Therefore, based on the need to prevent and treat falls, there has been an interest in modifiable and monitorable factors, such as physical activity and sarcopenia.

Although falls are becoming more common as older people age, they are not uncontrollable and can be prevented through targeted exercise interventions. International guidelines recommend that older adults participate in at least 150 min of moderate-intensity physical activity (PA) per week [10]. However, approximately 33.3% of the world’s population is failing to meet the minimum guidelines [11]. With almost one-fifth of the world’s population, it can be deduced that the Chinese are highly likely to suffer from physical inactivity. Data regarding physical activity in China is scarce and most studies are restricted in representativeness. Therefore, further research is necessary to not only determine the prevalence of physical inactivity, but also to investigate its correlation with falls so that effective interventions can be developed. Fortunately, the China Health and Retirement Longitudinal Study (CHARLS) provides a comprehensive national survey to address some of the current limitations in evaluating physical activity in China. This survey covers the majority of provinces, autonomous regions, and municipalities in mainland China (with the exception of Tibet), and a total of 21,095 people took part in the initial survey with face-to-face interviews and standardized questionnaires.

Sarcopenia, a syndrome that Rosenberg first mentioned in 1989, is a steady decline in muscle strength and mass as one ages [12], which can result in a range of unfavorable outcomes, such as decreased functioning, frailty, and death [13,14]. Sarcopenia may play a major role in falls among older adults. Several studies have indicated that physical inactivity has a great influence on the progression of sarcopenia [15,16], while participating in PA can help seniors to build muscle strength and mass [17]. In addition, individuals with sarcopenia are more susceptible to fall and fractures than those without it [7]. Accordingly, it is essential to assess the prevalence of sarcopenia in China and to identify individuals who may be suffering from it in order to enhance the well-being and quality of life of older adults.

Currently, most studies on the relationship between PA, sarcopenia, and falls are from Western countries. However, there are only a few studies that have been conducted on Chinese community residents. Additionally, some studies have used the evaluation criteria for western countries for Chinese people [18], but due to the different ethnicities, genetic backgrounds, and living environments, it would be more appropriate to use the Asian Working Group on Sarcopenia (AWGS) 2019 criteria for Chinese older adults. The purpose of this research is to analyze the mediating role of sarcopenia in the link between physical activity and falls among Chinese older adults.

In the research, we proposed two hypotheses regarding the mediating role of sarcopenia and its constituent factors in PA and falls among Chinese older adults. Hypothesis 1 suggests that sarcopenia and its components mediate the connection between PA and falls, while Hypothesis 2 proposes that PA mediates the association between sarcopenia and its components and falls.

## 2. Materials and Methods

### 2.1. Study Design and Participants

In total, 21,095 valid samples were gathered in 2015, of which 9871 were from people aged 60 and over. We excluded participants with missing data on PA, sarcopenia, and falls. The final sample analyzed, therefore, comprised 3592 individuals (Figure 1).

Due to the absence of the latest 2018 data from the CHARLS platform on physical examination questionnaires, this study employed the 2015 data to analyze the connection between physical activity, sarcopenia, and falls in Chinese senior citizens. CHARLS, sponsored by the National Development Research Institute at Peking University, utilized a multi-stage PPS sampling strategy to survey individuals of middle and advanced age living in 28 Chinese provinces, encompassing a total of 150 counties and 450 communities. This project is focused on gathering first-class data concerning multiple health aspects, including basic personal information, health status, physical measurements, health service usage and health insurance, work, and income, etc.

### 2.2. Assessment of PA

The CHARLS questionnaire classifies PA as vigorous, medium, and low. Vigorous exercise is described as making breathing much more difficult than normal and may include lifting heavy objects, digging, plowing, aerobics, fast cycling, and heavy cycling; moderate PA includes breathing a little harder than normal and may include lifting light objects, riding a bicycle at a normal speed, or mopping the floor; and low PA is walking—traveling from place to place, and any other walking undertaken solely for recreation, sport, exercise, or leisure.

In each activity category, the frequency per week (d/w) and the cumulative time per day (min/d) of the three different intensities were examined. During data cleaning, the principle of data truncation was followed [19]. If the daily time of a certain intensity PA exceeded 3 h, it would be recorded as 180 min. The principle allows an upper limit of 21 h (1260 min) per week of PA of each intensity to be reported. To estimate the utilization of energy for physical exercise in a week, the metabolic equivalent of each PA mode was taken into account [20]. The weekly PA energy expenditure was estimated for the metabolic equivalent (1) of each PA mode.
(1)MET-min/w=MET∗frequency(d/w)∗time(min/d)

The IPAQ short form assigns a MET of 3.3 for walking, 4.0 for moderate intensity activities, and 8.0 for vigorous intensity activities.These three intensity levels of PA add up to the total PA level. Lastly, the International Physical Activity Scale (IPAQ) divides the physical activity into three categories: low (<600 METs/wk), medium (600–3000 METs/wk), and high (>3000 METs/wk) [21].

### 2.3. Definition of Sarcopenia

The Asian Working Group on Sarcopenia (AWGS) 2019 guidelines have determined sarcopenia according to three components: muscle strength, physical performance, and appendicular skeletal muscle mass (ASM) [22]. Identification of sarcopenia is poor muscle mass or physical performance in conjunction with a low ASM. Possible sarcopenia is characterized by either low muscle strength or low physical function [23].

Handgrip strength (HGS) is a measure of muscle strength determined by the maximum force that can be applied when squeezing a dynamometer. Participants were instructed to give their utmost effort using their left hand, followed by their right. The mean of the most powerful HGS of the two hands was used as the HGS value; if the testers were unable to perform the test with one hand, the data from the other hand was selected as the HGS value. The cut-off for HGS for males was established at 28 kg, and for females, at 18 kg.

The five-time chair stand test (5-CST) is used to measure physical performance. The 5-CST requires participants to stand up straight and then sit down again at their fastest pace five times without stopping in between and without using arms to push off. If the 5-CST result is >12 s, it is labeled as having low physical performance.

ASM was calculated through a physical measurement Formula (2) verified in the Chinese population [24]. The gender is set to 1 if it is male, otherwise to 0. Previous studies have established a powerful connection between the ASM formula and dual X-ray absorptiometry (DXA) [25,26]. Women with an ASM/Ht2 value of <5.4 kg/m^2^ and men with an ASM/Ht2 value of <7.0 kg/m^2^ are thought to have low muscle mass [22].
(2)ASM = 0.193 ∗ weight (kg)+0.107 ∗ height (cm) − 4.157 ∗ gender − 0.037 ∗ age(years) − 2.631

### 2.4. Fall

The dependent variable in this paper is falls. Falls were self-reported by respondents as to whether they had fallen within two years. The question was: “Have you fallen since 2013?”. Those who said “yes” were considered to have fallen and given a value of 1, with 0 assigned to those who answered otherwise.

### 2.5. Covariates

Taking into account the findings of prior research, we chose socio-demographic and health-related factors for this research. These variables included gender, age, residence area (urban or town, rural), education (primary school or below, senior or high school, college or above), and marital status (married, unmarried/divorced/widowed). Additionally, sleep time, ever/current smoking (yes or no), ever/current alcohol consumption (yes or no), chronic disease (yes or no), and body mass index (BMI) were also taken into consideration.

### 2.6. Statistical Analysis

For categorical variables, numbers and percentages were used, while for continuous variables, the mean and standard deviation (SD) were reported. So as to explore the connection between PA, sarcopenia, and falls, three logistic regression models were employed to calculate the odds ratio (OR) and its corresponding 95% confidence interval (95% CI). Model 1 was the unaltered model, and Model 2 was modified to include gender, age, marital status, residential area, and education. Model 3 was built upon Model 2, taking into account sleep duration, smoking, alcohol consumption, chronic disease, and BMI.

Two sets of mediation models were constructed and analyzed separately. In a mediating model, we used PA as the dependent variable, sarcopenia and its components as mediating variables, and falls as the independent variable (Figure 2a). We also examined the mediating role of PA levels between sarcopenia and falls in another model (Figure 2b). To examine the mediating effect, Karlson, Holm and Breen’s method (KHB) was used to calculate the coefficient [27]. The KHB method is capable of separating out the impact of discrete and continuous variables, and broadening the capacity of linear models to encompass non-linear probabilistic models.

Gender, age, residence area, marital status, education, sleep time, smoking, drinking, chronic disease, and BMI were adjusted for in the KHB model. All data analysis was tested using Stata 17.0 software (StataCorp, College Station, TX, USA). Statistically significant results were determined if *p* < 0.05.

## 3. Results

### 3.1. Descriptive Statistics

A total of 3592 participants were included in the analysis (Table 1). Descriptive statistics of the participants, categorized by falls, are shown in Table 1. The participants consisted of 50.14% males and 49.86% females, and the mean age was 67.87 years. The fall rate was 20.02% and the prevalence of sarcopenia was 6.57%. We found that falls tended to occur in people who were older, female, living in rural areas, having lower levels of education, and having shorter sleep duration (*p* < 0.001). Among other key variables, low physical activity levels, sarcopenia, lower muscle strength, poorer physical function, and lower muscle mass, were significantly correlated with falls.

### 3.2. Association of PA, Sarcopenia, and Falls

To analyze the link between physical activity, sarcopenia, and falls, logistic regression analysis was used to calculate the OR with 95% CI. As shown in Table 2, we found that physical activity was a protective factor against falls in older adults (OR = 0.54, 95% CI = 0.48, 0.61), while sarcopenia was a hazard factor (OR = 1.34, 95% CI = 1.16, 1.55). The research Hypotheses 1 and 2 in this article were supported. Furthermore, components of sarcopenia, such as better HGS (OR = 0.96, 95% CI = 0.95, 0.98) and ASM (OR = 0.42, 95% CI = 0.30, 0.59), were associated with a lower risk of falls. Prolonged 5-CST time increased the chances of falls (OR = 1.07, 95% CI = 1.04, 1.10).

### 3.3. Mediating Effect of Sarcopenia between PA and Falls

As shown in Table 3, sarcopenia acted as a significant mediator in the connection between PA and falls (estimated indirect effect: −0.02, 95%CI: −0.03, −0.01, mediating effect proportion: 2.69%). Similarly, the mediating effects of HGS, 5-CST, and ASM on physical activity and falls were significant, with percentages of 4.14%, 4.61%, and 7%, respectively. As a result, hypothesis 3 of this study was validated.

### 3.4. Mediating Effect of PA between Sarcopenia and Falls

Our findings indicated that physical activity had a more powerful mediating influence on the association between sarcopenia and falls (estimated indirect effect: 0.06, 95%CI: 0.03, 0.09, mediating effect proportion: 20.12%). PA also played a significant mediating role in the associations between HGS, 5-CST, and ASM, with the proportions of the mediating effect being 14.94%, 18.75%, and 13.79%, respectively.

## 4. Discussion

The current cross-sectional study demonstrates that PA and sarcopenia are both linked to falls in Chinese older adults living in the community. Moreover, it was found that sarcopenia had a mediating effect on the connection between PA and falls. Furthermore, the relationship between sarcopenia and falls was also mediated by PA.

The fall prevalence of 20.02% among older adults is consistent with the previously reported rate of 20.83% [28], as well as a study on Chinese community old people which revealed a fall prevalence of 22.01%. Our findings support the earlier evidence that those who have experienced falls are generally older than those who have not [29,30]. The aging process can bring about a decrease in physical activity [31], cognitive function, motor neuron function, and skeletal muscle mass, as well as an increase in sarcopenia [32]. Compared to men, women’s rate of falling is almost twice as much. This phenomenon was also found in previous studies in the United States [33] and Canada [34] among seniors. This could be attributed to the fact that approximately 60% of falls occur within the home, rather than while engaging in outdoor activities. Living rooms, bedrooms and kitchens were the most frequent sites for indoor falls, and women were particularly prone to falls in the kitchen and while engaging in housework [35,36]. In addition, as the ovarian function of older women diminishes after menopause, the secretion of estrogen decreases [37], causing an imbalance in bone metabolism and greater bone loss [38], thus making them more prone to osteoporosis. Furthermore, with age, their bone density decreases and their muscle strength and body functions weaken, increasing their chances of falls and fractures [39].

Our research revealed that individuals with low physical activity were more than twice as likely to experience falls, which is in line with the notion that the less physical activity one engages in, the more likely one is to experience falls [40,41]. Nevertheless, some theories suggested that physical activity levels and falls have a U-shaped relationship [42,43]. This could be the result of individuals who are highly active in exercise exceeding their limits [44], while people with limited mobility may have a better grasp of their physical fitness, or exercise with the supervision of a doctor, due to their limited mobility. Hence, the level of activity is directly linked to their capabilities. However, men with normal physical function may be unable to take preventive steps against falls due to exhaustion, or may be more vulnerable to engaging in activities that are not suitable for their age and abilities [45].

Among Chinese seniors in the community, the probable rates of possible sarcopenia and sarcopenia were calculated to be 47.57% and 11.13%, and the rate of falls was greater among those with probable sarcopenia and sarcopenia than those without it. This conclusion is in agreement with studies conducted both domestically [46] and abroad [47]. However, Clynes et al. observed that there is no significant correlation between falls and sarcopenia, which could be attributed to the different definitions and measurements of sarcopenia [47,48]. Furthermore, our findings imply a significant reduction in falls when considering the components of sarcopenia. Possible sarcopenia is associated with a higher rate of falls, implying that poor muscle function and muscle strength are more significant contributors to falls than muscle mass. Similarly, the American Geriatrics Society and the British Geriatrics Society identify muscle strength and physical performance as potential hazards for falls [49]. Furthermore, a study conducted in Japan on 1,110 senior participants revealed that, in comparison to those who had not fallen, individuals who had fallen during the last year had weaker handgrip strength and physical function. However, the link between muscle mass and falls was uncertain [50].

The study found that sarcopenia significantly mediated the correlation between PA and falls. This implies that a lack of PA can cause sarcopenia, which then increases the risk of falls. Sarcopenia is responsible for 2.69% of the connection between PA and falls. Previous studies have revealed that physical inactivity is a primary factor for sarcopenia, and the prevalence of sarcopenia in those with physical inactivity is significantly higher than those with moderate to high PA [51]. Older adults may experience physical or psychological issues that lead to a reduction in their PA and social engagement, resulting in decreased muscle mass and physical performance due to inadequate exercise. Moreover, research conducted on Chinese individuals has revealed that PA and sarcopenia may have a reciprocal relationship [52]. Elderly individuals who engage in regular exercise have greater handgrip strength and physical function compared to those who do not, suggesting that exercise can boost physical capability and lower the risk of sarcopenia [53]. Consequently, it can be assumed that sarcopenia and its components are mediators between PA and falls.

Sarcopenia and PA both have a mediating effect on the association between one another and falls; there are numerous common mechanisms that can explain the pathways. First, chronic inflammation could potentially be the root cause. Our research showed that older people who had a fall were more likely to have a chronic illness. Older adults often suffer from multiple chronic conditions, and inflammatory cytokines can activate the ubiquitin proteasome pathway, resulting in the breakdown of muscle fibers [54], leading to a reduction in strength, function, and muscle mass, hastening the development of sarcopenia, which results in a greater chance of falls [55]. It appears that physical exercise has a restraining effect on inflammatory reactions [56]. Secondly, those who fell were found to have significantly shorter sleep duration than those who did not (6.69 ± 0.08 vs 7.14 ± 0.04, *p* < 0.001). Sleep disorder is a frequent cause of physical inactivity and sarcopenia [57]. Insufficient sleep can cause a reduction in insulin-like growth factors (IGF-1) and testosterone secretion [58], and an increase in cortisol levels [59]. A decrease in IGF-1 and testosterone levels results in a diminishment in the synthesis of muscle proteins, an increase in skeletal muscle proteolysis, and an increase in the expression of muscle growth inhibitors. At the same time, elevated cortisol levels can cause sarcopenia [60]. Thirdly, aging was associated with a notable rise in insulin resistance, which in turn had a negative impact on muscle strength among the older persons. Physical exercise not only directly enhances insulin sensitivity, but also indirectly adjusts body composition, thereby controlling muscle and liver insulin sensitivity [61] and delaying the emergence of sarcopenia.

## 5. Strengths and Limitations

Our article has certain strong points. Firstly, as far as we are aware, no other research has looked into the link between PA, sarcopenia, and falls using the CHARLS National Survey data, which was collected from a representative sample of Chinese older adults, and can be utilized for the whole elderly population in China. Secondly, we utilized the AWGS guidelines to measure sarcopenia, which is more compatible with the muscular characteristics of Asians. Additionally, we examined not only the correlation between PA, sarcopenia, and falls, but also assessed the role of sarcopenia and PA as mediators.

However, there are still certain caveats. Firstly, our survey was based on self-reported data regarding the duration, intensity and frequency of PA and falls, and this may not be reliable as some people may be inclined to deny or understate their issues. However, the accuracy of the measurements of these variables has been validated in prior studies [62]. Secondly, instead of relying on dual X-ray absorptiometry (DXA) or bioelectrical impedance analysis (BIA), which are endorsed by the AWGS 2019, we opted to use anthropometric equations that have been validated in Chinese studies. Thirdly, as ours was a cross-sectional study, we were unable to acquire more in-depth mediating information. To determine causality in relation to pathways, we will carry out further longitudinal studies utilizing the most recent information from the CHARLS 2024 survey to provide us a more profound understanding [63].

## 6. Conclusions

In conclusion, this study demonstrates a negative correlation between PA and falls in Chinese older adults, and that this relationship is partly transmitted through sarcopenia. This implies that, by improving physical function, muscle strength, and muscle mass in people with sarcopenia, the incidence of falls may be lowered. Given these findings, physical function and muscle strength should be regularly assessed and tested during community screenings. In addition, healthcare providers should focus on older adults who are less physically active and provide individualized exercise programs to help them participate in physical activities.

## Figures and Tables

**Figure 1 healthcare-11-03146-f001:**
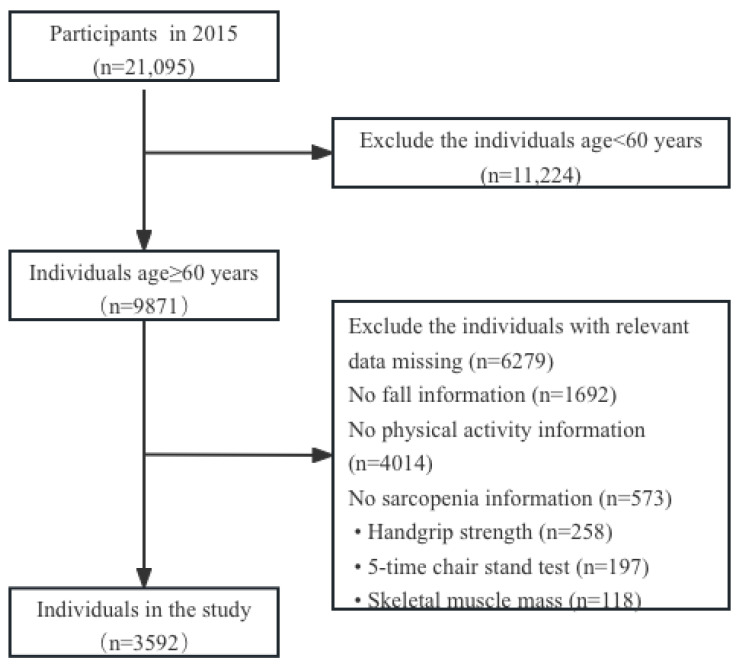
Flowchart illustrating the screening process.

**Figure 2 healthcare-11-03146-f002:**
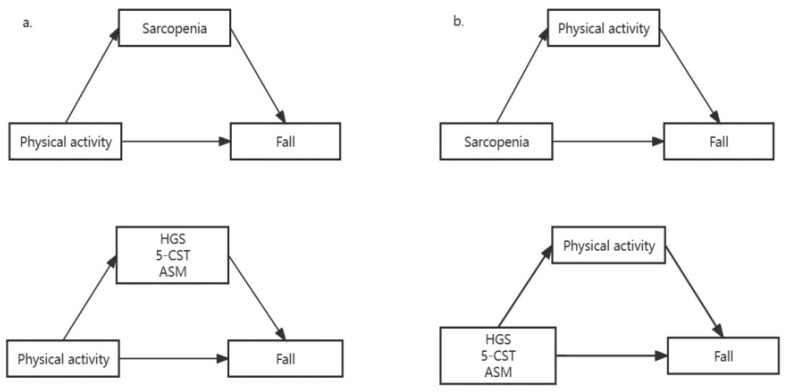
Mediation models of physical activity, sarcopenia, including its components, and falls in Chinese older adults. Panel (**a**) illustrates the potential mediating effect of sarcopenia and its components on the correlation between physical activity and falls. Panel (**b**) implies that physical activity may act as a mediator between sarcopenia, including its components and falls.

**Table 1 healthcare-11-03146-t001:** Characteristics of participants.

Variables	Total N = 3592	Fall N = 719 (20.02)	Non-Fall N = 2873 (79.98)	*p* Value
Gender				<0.001
Male	1801 (50.14)	272 (37.83)	1529 (53.22)	
Female	1791 (49.86)	447 (62.17)	1344 (46.78)	
Age (years)	67.87 ± 6.11	68.20 ± 0.24	67.71 ± 0.12	
Resident area				<0.001
Urban or town	809 (22.52)	130 (18.08)	679 (23.63)	
Rural	2783 (77.48)	589 (81.92)	2194 (76.37)	
Education				<0.001
Primary school or below	2460 (68.49)	540 (75.10)	1920 (66.83)	
Senior or high school	816 (22.72)	133 (18.50)	683 (23.77)	
College or above	316 (8.80)	46 (6.40)	270 (9.40)	
Marital status				0.004
Married	2923 (81.38)	558 (77.61)	2365 (82.32)	
Never-married/ Separated/widowed	669 (18.62)	161 (22.39)	508 (17.68)	
Sleep time (h/d)	7.04 ± 2.28	6.69 ± 0.08	7.14 ± 0.04	<0.001
Ever/current smoke				<0.001
Yes	792 (22.05)	252 (35.05)	1335 (46.47)	
No	2800 (77.95)	467 (64.95)	1538 (53.53)	
Ever/current alcohol				0.013
Yes	1173 (32.66)	207 (28.79)	966 (33.62)	
No	2419 (67.34)	512 (71.21)	1907 (66.38)	
Chronic disease				<0.001
Yes	778 (21.66)	201 (27.96)	577 (20.09)	
No	2814 (78.34)	518 (72.04)	2296 (79.91)	
BMI (kg/m^2^)	23.44 ± 4.02	23.67 ± 0.14	23.26 ± 0.72	0.971
Physical activity				<0.001
LPA	532 (14.81)	209 (29.07)	323 (11.24)	
MPA	1527 (42.51)	271 (37.69)	1256 (43.72)	
VPA	1533 (42.68)	239 (33.24)	1294 (45.04)	
Sarcopenia				<0.001
Non-sarcopenia	2173 (60.50)	277 (38.53)	1896 (65.99)	
Possible sarcopenia	1183 (32.93)	342 (47.57)	841 (28.27)	
Sarcopenia	236 (6.57)	80 (11.13)	156 (5.43)	
HGS (kg)	27.61 ± 0.15	24.94 ± 0.31	28.28 ± 0.16	<0.001
5-CST (s)	10.24 ± 0.07	11.07 ± 0.16	10.03 ± 0.62	<0.001
ASM (kg/m^2^)	8.45 ± 1.77	8.22 ± 0.05	8.50 ± 0.02	<0.001

Note: The data are displayed as mean ± SD and number (percentage); BMI, body mass index; LPA, low physical activity; MPA, moderate physical activity; VPA, vigorous physical activity; HGS, handgrip strength; 5-CST, five-time chair stand test; ASM, appendicular skeletal muscle.

**Table 2 healthcare-11-03146-t002:** Logistic regression analysis of PA, sarcopenia, and falls.

Variables	Model 1 OR (95% CI)	*p* Value	Model 2 OR (95% CI)	*p* Value	Model 3 OR (95% CI)	*p* Value
Physical activities	0.56 (0.50, 0.63)	<0.001	0.55 (0.48, 0.62)	<0.001	0.54 (0.48, 0.61)	<0.001
Sarcopenia	1.30 (1.15, 1.48)	<0.001	1.32 (1.14, 1.52)	<0.001	1.34 (1.16, 1.55)	<0.001
HGS	0.95 (0.94, 0.97)	<0.001	0.96 (0.95, 0.97)	<0.001	0.96 (0.95, 0.98)	<0.001
5-CST	1.08 (1.05, 1.10)	<0.001	1.07 (1.04, 1.09)	<0.001	1.07 (1.04, 1.10)	<0.001
ASM	1.15 (1.08, 1.23)	0.02	0.91 (0.82, 1.02)	<0.001	0.42 (0.30, 0.59)	<0.001

Note: Model 1 was the original model; Model 2 was adjusted for age, gender, residential area, education, and marital status; Model 3 was further adjusted to account for all the covariates.

**Table 3 healthcare-11-03146-t003:** Mediation effects of PA and sarcopenia on falls.

*IV*	M	DV	Total Effect	Direct Effect	Indirect Effect	Mediating Effect (%)	*p* Value
PA	Sarcopenia	Fall	−0.61 (−0.73, −0.49)	−0.60 (−0.72, −0.48)	−0.02 (−0.03, −0.01)	2.69	<0.001
	HGS		−0.62 (−0.74, −0.50)	−0.59 (−0.71, −0.47)	−0.03 (−0.04, −0.01)	4.14	<0.001
	5-CST		−0.63 (−0.75, −0.50)	−0.59 (−0.71, −0.47)	−0.03 (−0.04, −0.01)	4.61	<0.05
	ASM		−0.62 (−0.75, −0.49)	−0.58 (−0.70, −0.46)	−0.04 (−0.07, −0.02)	7.00	<0.001
Sarcopenia	PA		0.30 (0.15, 0.44)	0.24 (0.09, 0.38)	0.06 (0.03, 0.09)	20.12	<0.001
HGS			−0.04 (−0.05, −0.02)	−0.03 (−0.05, 0.02)	−0.01 (−0.01, 0.00)	14.94	<0.05
5-CST			0.07 ( 0.04, 0.09)	0.06 (0.03, 0.08)	0.01 (0.01, 0.02)	18.75	<0.001
ASM			−0.89 (−1.24, −0.53)	−0.77 (−1.12, −0.41)	−0.12 (−0.17, −0.08)	13.79	<0.001

Note: P-values and confidence intervals were calculated from the KHB method by Karson, Holm, and Breen (2011); all data are presented with a 95% confidence interval (95%CI); if 0 is not included in the 95% confidence interval, the mediating effect is deemed to be significant; all models were adjusted for all covariates; IV, independent variable; M, mediator; DV, dependent variable; PA, physical activity; 5-CST, five-time chair stand test; ASM, appendicular skeletal muscle.

## Data Availability

The CHARLS database holds raw data that is accessible to the public upon request. The website of CHARLS is URL http://charls.pku.edu.cn/ (accessed on 8 November 2022), and the datasets used and analysed in this study can be obtained by contacting the corresponding author.

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
