# Peer review of "The Mediating Role of Sarcopenia in the Association between Physical Activity and Falls among Chinese Older Adults: A Cross-Sectional Study"

_healthcare, 2023, doi:10.3390/healthcare11243146_

Round 1

Reviewer 1 Report

Comments and Suggestions for Authors

1)     I do not see the rationale for capitalization in the title.

2)     The final few sentences of the ABSTRACT are awkwardly written.  

3)     “It” statements (eg, “It is recommended by…”) should be eschewed- particularly if contributing to passive voice.

4)     The numbers in the flowchart are inconsistent. Some are in parentheses whereas others begin cell entrees.

5)     What was the basis of the sit-to-stand cut-off.

6)     In the Tables and elsewhere, I would think one decimal place would sufficient as a  summary .

7)     References are presented inconsistently.

Comments on the Quality of English Language

OK with minor needs for grammatical modifications. 

Author Response

Dear reviewer:

Re: Manuscript ID: ijerph-2598728 and Title: To What Degree does sarcopenia mediate the link between physical activity and falls? Evidence from the Longitudinal Study in China

Thank you for your letter and the reviewers’ comments concerning our manuscript. Those comments are valuable and very helpful. We have read through comments carefully and have made corrections. Based on the instructions, we uploaded the file of the revised manuscript. The responses to the reviewer’s comments are presented following.

We would love to thank you for allowing us to resubmit a revised copy of the manuscript and we highly appreciate your time and consideration.

Sincerely,

Lei Shi, Chenyu Liang, Baocheng Li, and Zhiyu HE

Reviewer #1:

Comments 1:I do not see the rationale for capitalization in the title.

Response 1:We are grateful for the suggestion. We've made a correction.

Comments 2:The final few sentences of the ABSTRACT are awkwardly written.  

Response 2:Thank you for your advice. The last few sentences of the ABSTRACT we have changed to express our findings more clearly and logically.

Comments 3: “It” statements (eg, “It is recommended by…”) should be eschewed- particularly if contributing to passive voice.

Response 3:We apologize for the language problems in the original manuscript. The language presentation was improved.

Comments 4: The numbers in the flowchart are inconsistent. Some are in parentheses whereas others begin cell entrees.

Response 4: Thank you very much for your suggestions. We agree with the comment and have redone the diagrams to ensure consistent formatting.

Comments 5: What was the basis of the sit-to-stand cut-off.

Response 5: The basis of the sit-to-stand cut off is based on the Asian Working Group on Sarcopenia (AWGS) 2019 guidelines mentioned in the article "2.3 Definition of Sarcopenia".

Comments 6: In the Tables and elsewhere, I would think one decimal place sufficient as a   summary.

Response 6: Thank you very much for your advice. We reviewed the papers and saw that most of them kept two decimal places, so we also kept all the data in two decimal places.

Comments 7: References are presented inconsistently.

Response 7: We apologize for the formatting problems with the references in the original manuscript. The correction has now been made.

Reviewer 2 Report

Comments and Suggestions for Authors

The purpose of the current study was to examine the associations between physical activity, sarcopenia, and factors that make up sarcopenia including muscle strength, physical performance, and muscle mass. The data from a total of 3592 seniors over the age of 60 was used for the study. The measures of sarcopenia, physical activity, muscle mass, strength and physical performance above were taken using standard measurements for each variable that are generally well-described in the literature (e.g. handgrip strength has been shown to be the best predictor of longevity). Logistic regression was used to examine the above associations. Accordingly, the main question addressed by the research was the associations between various factors that make up sarcopenia and the incidence of falls. This is important as falls are a major factor in death and medical costs in older adults. Although the topic is not extremely novel due to sarcopenia being studied for many years, the paper adds to the literature due to its high sample size and being conducted in a non-North American or European country in a different population and culture.

 The main findings were: 1) lower physical activity was associated with an increase in the likelihood of falls; 2) sarcopenia was positively associated with falls; 3) the components of sarcopenia were also each associated as falls. I think most people familiar with these topics would not be surprised by these findings, but it is good to have it quantified and evaluated in a large sample like the current study.

 Overall, this seems to be a well-done study on a basic, but very important topic in my opinion. The sample size was high and most of the measurements use are well-described in the literature. The analysis of the data appears to have no fatal flaws. I liked that the number of men and women was almost equal and that a large part of the sample was from rural areas. It was also done in China which provides additional information to the literature as most of these type of studies in the past have been in the USA or northern Europe.

I don’t have any major issues with the paper. Minor issues are that there are many instances of very minor English mistakes or awkward wording. There are also some typographical errors like lack of spaces in some instances (e.g. line 193 there is no space after the brackets in that line). I think another thing that would help the paper is to present some of the major and most important variables in figures to supplement the tables. This should allow the reader to better visualize the data. Finally, the reference list is appropriate but has some mistakes. Sometimes the journal name is spelled all the way out, other times it is abbreviated. In addition, the capitalization of the title of the journals is inconsistent. As one example just look at the differences in these aspects between references 65-70 within that sample of 6 of them. So I think the paper would be potentially publishable after moderate English and other proofreading is done along with some changes to the data presentation (e.g. data figures).

Comments on the Quality of English Language

See comments to authors

Author Response

Dear reviewer:

Re: Manuscript ID: ijerph-2598728 and Title: To What Degree does sarcopenia mediate the link between physical activity and falls? Evidence from the Longitudinal Study in China

Thank you for your precious comments and advice. Those comments are all valuable and very helpful for revising and improving our paper. We have revised the manuscript accordingly,and our point-by-point responses are presented.

We would love to thank you for allowing us to resubmit a revised copy of the manuscript and we highly appreciate your time and consideration.

Sincerely,

Lei Shi, Chenyu Liang, Baocheng Li, and Zhiyu HE

Comments 1:There are also some typographical errors like lack of spaces in some instances (e.g. line 193 there is no space after the brackets in that line)

Response 1:We are grateful for the suggestion. We've made a correction.

Comments 2:I think another thing that would help the paper is to present some of the major and most important variables in figures to supplement the tables.

Response 2:Thank you for your advice. In the results section, we add explanations of the results through figures.

Comments 3: The reference list is appropriate but has some mistakes. 

Response 3: We apologize for the formatting problems with the references in the original manuscript. The correction has now been made.

Reviewer 3 Report

Comments and Suggestions for Authors

The manuscript have several flats that need to correct:

Abstract

The abstract needs improvement in terms of clarity and structure. It should follow a more logical flow, starting with the problem statement, followed by the methodology and key findings. The opening sentence is vague. What exactly were you trying to investigate or determine?

Introduction

Consider restructuring the introduction to follow a clear pattern: problem statement, importance, relevance, and a brief overview of the study's objectives or hypotheses

While you highlight the importance of falls among the elderly, it would be helpful to explicitly state the research problem or question your study addresses

Emphasize the global relevance of the issue. You start with a broad statement about falls being a global problem, but then focus on China. Clarify whether your study is specific to China or if it has broader implications

The transition from discussing falls to introducing physical activity and sarcopenia could be smoother

Explicitly state what gap in knowledge or unanswered question your study aims to address concerning the relationship between falls, physical activity, and sarcopenia.

The materials and methods section provides a detailed description of the study design and procedures, which is commendable. However, some parts could benefit from improved clarity:

Explain why you are using data from 2015 to a 2023 study

While you describe the physical activity categories (high, medium, low), it would be helpful to provide the specific criteria or questions used to classify individuals into these categories in CHARLS

 Describe the formula or method used to calculate MET for each physical activity mode, as this affects the estimation of energy expenditure.

Explain how muscle strength, physical performance, and appendicular skeletal muscle mass (ASM) were assessed more clearly. Provide the cut-off values used to define sarcopenia

Are you using 2.5 or 6m for gait speed test? Please clarify and justify.

Describe the statistical methods and models used in more detail,

Results:

Tables are duplicate

provide p-values for the mediating effects to indicate their statistical significance

Discussion:

While you mention associations and mediating effects, provide more detailed explanations for these findings. For example, why does sarcopenia mediate the relationship between physical activity and falls? Offer hypotheses or mechanisms to help readers understand the underlying reasons.

Offer more detailed interpretations of the mediating effects. Discuss the practical implications of these mediation relationships.

While you touch on some potential biological mechanisms (e.g., chronic inflammation, sleep), consider expanding on these explanations with references to relevant studies. This will strengthen the scientific basis of your arguments

Be cautious of repeating points unnecessarily. For instance, you mention the relationship between sarcopenia, physical activity, and falls multiple times; try to consolidate these discussions into a coherent narrative.

it's important to thoroughly address the limitations associated with self-reported data, anthropometric equations, the cross-sectional design, and the need for more comprehensive mediation information. Discussing potential solutions and the generalizability of findings would further enhance the study's quality and impact

Conclusion:

It could be improved by providing more specific effect sizes, avoiding redundancy, clarifying the nature of causality, and offering concrete recommendations for future research. Additionally, adding a final thought could enhance the overall impact of the conclusion.

Comments on the Quality of English Language

Proofread for grammatical accuracy and clarity. Please avoid “elderly”. It is older adults

Author Response

Dear reviewer:

Re: Manuscript ID: ijerph-2598728 and Title: To What Degree does sarcopenia mediate the link between physical activity and falls? Evidence from the Longitudinal Study in China

Thank you for your comments concerning our manuscript. Those comments are valuable and very helpful. We have read through comments carefully and have made corrections. Based on the instructions, we uploaded the file of the revised manuscript. The responses to the reviewer’s comments are presented following.

We would love to thank you for allowing us to resubmit a revised copy of the manuscript and we highly appreciate your time and consideration.

Sincerely,

Lei Shi, Chenyu Liang, Baocheng Li, and Zhiyu HE

Comments 1:The abstract needs improvement in terms of clarity and structure. It should follow a more logical flow, starting with the problem statement, followed by the methodology and key findings. The opening sentence is vague. What exactly were you trying to investigate or determine?

Response 1:We are grateful for your suggestions. We have rewritten the summary section as you suggested to make it more logically and the problem statement and key findings are clearly written.

Comments 2:Consider restructuring the introduction to follow a clear pattern: problem statement, importance, relevance, and a brief overview of the study's objectives or hypotheses

Response 2:Thank you for your precious comments and advice. We have rewritten the introduction section according to the model you suggested and added the research hypotheses.

Comments 3: While you highlight the importance of falls among the elderly, it would be helpful to explicitly state the research problem or question your study addresses.

Response 3:We are grateful for your suggestions. In the introduction section, we add current research gaps and our research questions.

Comments 4: Emphasize the global relevance of the issue. You start with a broad statement about falls being a global problem, but then focus on China. Clarify whether your study is specific to China or if it has broader implications.

Response 4: Thank you very much for your suggestions. We agree with the comment and have rewritten the section. We have narrowed down the scope of our study and emphasized that our study is focused on the elderly in China.

Comments 5: The transition from discussing falls to introducing physical activity and sarcopenia could be smoother.

Response 5: Thank you very much for your suggestions. We added transition sentences at the beginning of the second and third paragraphs, respectively.

Comments 6: Explicitly state what gap in knowledge or unanswered question your study aims to address concerning the relationship between falls, physical activity, and sarcopenia.

Response 6: Thank you very much for your advice. In the fourth paragraph we describe the current research gap and lead along to our research hypotheses.

Comments 7: Explain why you are using data from 2015 to a 2023 study.

Response 7: We apologize for mentioning this problem. The latest data (2018) currently in the CHARLS database is missing the medical examination questionnaire section, so we are unable to obtain the data we want. The next installment of data that was supposed to be released in 2022 has also been delayed due to the COVID-19 and will collect in 2023, so we will not be able to get the new data until 2024. Also, I explained the reason in the opening section of article 2.1.

Comments 8: While you describe the physical activity categories (high, medium, low), it would be helpful to provide the specific criteria or questions used to classify individuals into these categories in CHARLS.

Response 8: Thank you for your valuable advice. I've added specific criteria for dividing physical activity levels in the CHARLS questionnaire in section 2.2 of the article.

Comments 9: Describe the formula or method used to calculate MET for each physical activity mode, as this affects the estimation of energy expenditure.

Response 9: We are grateful for your suggestions. We have added Equation (1) in Section 2.2 of the article, which is used to help the reader understand how MET is calculated more clearly.

Comments 10: Explain how muscle strength, physical performance, and appendicular skeletal muscle mass (ASM) were assessed more clearly. Provide the cut-off values used to define sarcopenia.

Response 10: Thank you for your useful comments. We talk in more detail about how the sarcopenia is measured and what the cut-off values is in section 2.3 of the article.

Comments 11: Are you using 2.5 or 6m for gait speed test? Please clarify and justify.

Response 11: Thanks for your questions and suggestions. We don't use the gait speed test. According to AWGS guidelines, we use the five-time chair stand test (5-CST) to measure physical performance.

Comments 12: Tables are duplicate.

Response 12: We apologize for the problems in the original manuscript. The presentation was improved. We've made a correction.

Comments 13: provide p-values for the mediating effects to indicate their statistical significance.

Response 13: Thank you for your valuable advice. The p-value was added to the table of mediating effects.

Comments 14: While you mention associations and mediating effects, provide more detailed explanations for these findings. For example, why does sarcopenia mediate the relationship between physical activity and falls? Offer hypotheses or mechanisms to help readers understand the underlying reasons.

Response 14: We are grateful for your suggestions. We have added hypotheses and explained the findings in more detail in the discussion.

Comments 15: Offer more detailed interpretations of the mediating effects. Discuss the practical implications of these mediation relationships.

Response 15: Thank you for your useful comments. During the discussion, we tried to explain the mediating effect, however, it may not have been up to your expectations. We would be grateful if you could provide us with some constructive feedback.

Comments 16: Be cautious of repeating points unnecessarily. For instance, you mention the relationship between sarcopenia, physical activity, and falls multiple times; try to consolidate these discussions into a coherent narrative.

Response 16: We apologize for the language problems in the original manuscript. Following your suggestion, we have revised the language redundancy section to make the language more concise.

Comments 17: it's important to thoroughly address the limitations associated with self-reported data, anthropometric equations, the cross-sectional design, and the need for more comprehensive mediation information. Discussing potential solutions and the generalizability of findings would further enhance the study's quality and impact.

Response 17: Thank you for your valuable advice. The self-reported data and anthropometric equations certainly have some drawbacks. We confirmed the reliability of the data by reviewing and citing relevant literature. We also hope to do a tracking study to fill in the gaps of cross-sectional design after the latest outing is released.

Comments 18: It could be improved by providing more specific effect sizes, avoiding redundancy, clarifying the nature of causality, and offering concrete recommendations for future research. Additionally, adding a final thought could enhance the overall impact of the conclusion.

Response 18: Thank you for your useful comments. We have rewritten the conclusion section according to the form you suggested, and we hope you will give us your valuable suggestions!

Comments 19: Proofread for grammatical accuracy and clarity. Please avoid “elderly”. It is older adults.

Response 19: We apologize for the language problems in the original manuscript. The language presentation was improved.

We sincerely ask you to provide some valuable comments on our articles and we will try to correct them.

Round 2

Reviewer 3 Report

Comments and Suggestions for Authors

Thank you for resubmitting your paper to the journal. I have read it carefully and, despite the many improved done, I have few comments/suggestions for improvement:

- With this new version, the title of the paper is a little vague. It does not clearly state the research question, the population, the intervention, and the outcome. I suggest you revise it to something more concise and specific, such as “The mediating role of sarcopenia in the association between physical activity and falls among older adults in China: a cross-sectional study”

- Currently, your introduction still does not present a clear research gap or question.

- your conclusion section is too long and does not highlight the key messages of your paper. Please, be more specific with the take-home message.

With these little changes, I think the manuscript will be ready to publish

Congratulations for your work
